# Neural ODE Processes:
# A Short Summary

**Alexander Norcliffe**[*]
University of Cambridge
Cambridge, United Kingdom
alin2@cam.ac.uk

**Cristian Bodnar**[*]
University of Cambridge
Cambridge, United Kingdom
cb2015@cam.ac.uk

**Ben Day**[*]
University of Cambridge
Cambridge, United Kingdom
bjd39@cam.ac.uk

**Jacob Moss**[*]
University of Cambridge
Cambridge, United Kingdom
jm2311@cam.ac.uk

**Pietro Liò**
University of Cambridge
Cambridge, United Kingdom
pl219@cam.ac.uk

## Abstract

Neural Ordinary Differential Equations (NODEs) use a neural network to model the instantaneous rate of change in the state of a system. However, despite their apparent suitability for dynamics-governed time-series, NODEs present a few disadvantages. First, they are unable to adapt to incoming data-points, a fundamental requirement for real-time applications imposed by the natural direction of time. Second, time-series are often composed of a sparse set of measurements, which could be explained by many possible underlying dynamics. NODEs do not capture this uncertainty. To this end, we introduce Neural ODE Processes (NDPs), a new class of stochastic processes determined by a distribution over Neural ODEs. By maintaining an adaptive data-dependent distribution over the underlying ODE, we show that our model can successfully capture the dynamics of low-dimensional systems from just a few data-points. At the same time, we demonstrate that NDPs scale up to challenging high-dimensional time-series with unknown latent dynamics such as rotating MNIST digits. Code is available online at https://github.com/crisbodnar/ndp.

## 1  Background and Formal Problem Statement

**Problem Statement**  We consider modelling random functions $F : \mathcal{T} \to \mathcal{Y}$, where $\mathcal{T} = [t_0, \infty)$ represents time and $\mathcal{Y} \subset \mathbb{R}^d$ is a compact subset of $\mathbb{R}^d$. We assume $F$ has a distribution $\mathcal{D}$, induced by another distribution $\mathcal{D}'$ over some underlying dynamics that govern the time-series. For example, filming a pendulum, $F$ has a distribution in pixel space which is induced by the distribution of the pendulum dynamics. Given a specific instantation $\mathcal{F}$ of $F$, let $C = \{(t_i^C, \boldsymbol{y}_i^C)\}_{i \in I_C}$ be a set of samples from $\mathcal{F}$ with some indexing set $I_C$. We refer to $C$ as the context points, as denoted by the superscript C. For a given context $C$, the task is to predict the values $\{\boldsymbol{y}_j^T\}_{j \in I_T}$ that $\mathcal{F}$ takes at a set of target times $\{t_j^T\}_{j \in I_T}$, where $I_T$ is another index set. We call $T = \{(t_j^T, \boldsymbol{y}_j^T)\}$ the target set. Additionally let $t_C = \{t_i | i \in I_C\}$ and similarly define $y_C$, $t_T$ and $y_T$. Conventionally, as in Garnelo et al. [5], the target set forms a superset of the context set and we have $C \subseteq T$. Optionally, it might also be natural to consider that the initial time and observation $(t_0, \boldsymbol{y}_0)$ are always included in C. During training, we let the model learn from a dataset of (potentially irregular) time-series sampled from $F$. We are interested in learning the underlying distribution over the dynamics as well as the induced distribution

---

[*]Equal contribution.

Workshop Paper at The Symbiosis of Deep Learning and Differential Equations Workshop at NeurIPS 2021

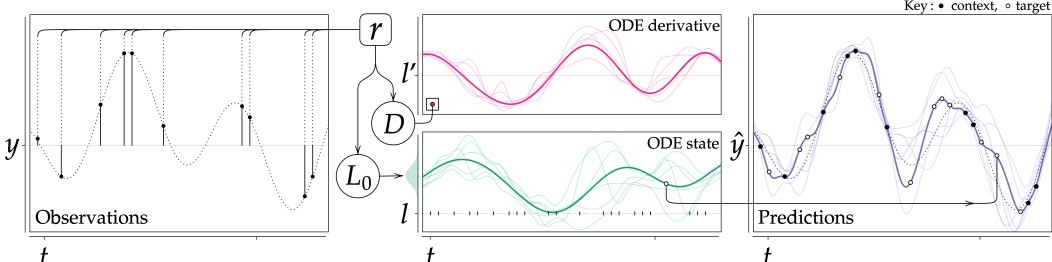

Figure 1: Schematic diagram of Neural ODE Processes. *Left:* Observations from a time series, the context set ●, are encoded and aggregated to form $r$ which parametrises the latent variables $D$ and $L_0$. *Middle:* A sample is drawn from $L_0$ and $D$, initialising and conditioning the ODE, respectively. Each sample produces a plausible, coherent trajectory. *Right:* Predictions at a target time, $t_i^{\mathrm{T}}$, are made by decoding the state of the ODE, $l(t_i^{\mathrm{T}})$ together with $t_i^{\mathrm{T}}$. An example is shown with the ○ connected from the ODE position plot to the Predictions plot. *Middle & right:* the bold lines in each plot refer to the same sample, fainter lines to other samples. *All:* The plots are illustrations only.

over functions. We note that when the dynamics are not latent and manifest directly in the observation space $\mathcal{Y}$, the distribution over ODE trajectories and the distribution over functions coincide.

**Neural ODEs** NODEs are a class of models that parametrize the velocity $\dot{z}$ of a state $z$ with the help of a neural network $\dot{z} = f_\theta(z, t)$. Given the initial time $t_0$ and target time $t_i^{\mathrm{T}}$, NODEs predict the corresponding state $\hat{y}_i^{\mathrm{T}}$ by performing the following integration and decoding operations:

$$z(t_0) = h_1(y_0), \qquad z(t_i^{\mathrm{T}}) = z(t_0) + \int_{t_0}^{t_i^{\mathrm{T}}} f_\theta(z(t), t) dt, \qquad \hat{y}_i^{\mathrm{T}} = h_2(z(t_i^{\mathrm{T}})), \tag{1}$$

where $h_1$ and $h_2$ can be neural networks. When the dimensionality of $z$ is greater than that of $y$ and $h_1, h_2$ are linear, the resulting model is an Augmented Neural ODE [4] with input layer augmentation [7]. The extra dimensions offer the model additional flexibility as well as the ability to learn higher-order dynamics [8].

**Neural Processes (NPs)** NPs [5] model a random function $F : \mathcal{T} \to \mathcal{Y}$, where in general $\mathcal{T} \subseteq \mathbb{R}^{d_1}$ and $\mathcal{Y} \subseteq \mathbb{R}^{d_2}$. The NP represents a given instantiation $\mathcal{F}$ of $F$ through the global latent variable $z$, which parametrises the variation in $F$. Thus, we have $\mathcal{F}(t_i) = g(t_i, z)$. For a given context set $\mathrm{C} = \{(t_i^{\mathrm{C}}, y_i^{\mathrm{C}})\}$ and target set $t_{1:n}, y_{1:n}$, the generative process is given by:

$$p(y_{1:n}, z | t_{1:n}, \mathrm{C}) = p(z|\mathrm{C}) \prod_{i=1}^{n} \mathcal{N}(y_i | g(t_i, z), \sigma^2), \tag{2}$$

where $p(z)$ is chosen to be a multivariate standard normal distribution and $y_{1:n}$ is a shorthand for the sequence $(y_1, \ldots, y_n)$. The model can be trained using an amortised variational inference procedure that naturally gives rise to a *permutation-invariant encoder* $q_\theta(z|\mathrm{C})$, which stores the information about the context points. Conditioned on this information, the *decoder* $g(t, z)$ can make predictions at any input location $t$. We note that while the domain $\mathcal{T}$ of the random function $F$ is arbitrary, in this work we are interested only in stochastic functions with domain on the real line (time-series). Therefore, from here we use a scalar variable $t$, instead of $t$. The output $y$ remains the same.

## 2 Neural ODE Processes

**Model Overview** We introduce Neural ODE Processes (NDPs), a class of dynamics-based models that learn to approximate random functions defined over time. To that end, we consider an NP whose context is used to determine a distribution over ODEs. Concretely, the context infers a distribution over the initial position (and optionally – the initial velocity) and, at the same time, stochastically controls its derivative function. The positions given by the ODE trajectories at any time $t_i^{\mathrm{T}}$ are then decoded to give the predictions. In what follows, we offer a detailed description of each component of the model. A schematic of the model can be seen in Figure 1.

## 2.1 Generative Process

We first describe the generative process behind NDPs. A graphical model perspective of this process is also included in Figure 2.

**Encoder and Aggregator** Consider a given context set $C = \{(t_i^C, y_i^C)\}_{i \in I_C}$ of observed points. We encode this context into two latent variables $L_0 \sim q_L(l(t_0)|C)$ and $D \sim q_D(d|C)$, representing the *initial state* and the *global control* of an ODE, respectively. To parametrise the distribution of the latter variable, the NDP encoder produces a representation $r_i = f_e((t_i^C, y_i^C))$ for each context pair $(t_i^C, y_i^C)$. The function $f_e$ is as a neural network, fully connected or convolutional, depending on the nature of $y$. An aggregator combines all the representations $r_i$ to form a global representation, $r$, that parametrises the distribution of the global latent context, $D \sim q_D(d|C) = \mathcal{N}\big(d|\mu_D(r), \mathrm{diag}(\sigma_D(r))\big)$. As the aggregator must preserve order invariance, we choose to take the element-wise mean. The distribution of $L_0$ might be parametrised identically as a function of the whole context by $q_L(l(t_0)|C)$, and, in particular, if the initial observation $y_0$ is always known, then $q_L(l(t_0)|C) = q_L(l(t_0)|y_0) = \mathcal{N}\big(l(t_0)|\mu_L(y_0), \mathrm{diag}(\sigma_L(y_0))\big)$.

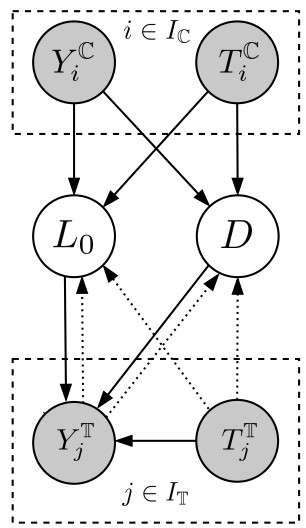

Figure 2: Graphical model of NDPs. The dark nodes denote observed random variables, while the light nodes denote hidden random variables. $I_C$ and $I_T$ represent the indexing sets for the context and target points, respectively. Full arrows show the generative process. Dotted arrows indicate inference.

**Latent ODE** To obtain a distribution over functions, we are interested in capturing the dynamics that govern the time-series and exploiting the temporal nature of the data. To that end, we allow the latent context to evolve according to a Neural ODE [2] with initial position $L_0$ and controlled by $D$. These two random variables factorise the uncertainty in the underlying dynamics into an uncertainty over the initial conditions (given by $L_0$) and an uncertainty over the ODE derivative, given by $D$.

By using the target times, $t_{1:n}^T = (t_1^T, ..., t_N^T)$, the latent state at a given time is found by evolving a Neural ODE:

$$l(t_i^T) = l(t_0) + \int_{t_0}^{t_i^T} f_\theta(l(t), d, t)dt, \tag{3}$$

where $f_\theta$ is a neural network that models the derivative of $l$. As explained above, we allow $d$ to modulate the derivative of this ODE by acting as a global control signal. Ultimately, for fixed initial conditions, this results in an uncertainty over the ODE trajectories.

**Decoder** To obtain a prediction at a time $t_i^T$, we decode the latent state of the ODE at time $t_i^T$, given by $l(t_i^T)$. Assuming that the outputs are noisy, for a given sample $l(t_i^T)$ from this stochastic state, the decoder $g$ produces a distribution over $Y_{t_i}^T \sim p\big(y_i^T|g(l(t_i^T), t_i^T)\big)$ parametrised by the decoder output. Concretely, for regression tasks, we take the target output to be normally distributed with constant (or optionally learned) variance $Y_{t_i}^T \sim \mathcal{N}\big(y_i^T|g(l(t_i^T), t_i^T), \sigma^2\big)$. When $Y_{t_i}^T$ is a random vector formed of independent binary random variables (e.g. a black and white image), we use a Bernoulli distribution $Y_{t_i}^T \sim \prod_{j=1}^{\dim(Y)} \mathrm{Bernoulli}\big(g(l(t_i^T), t_i^T)_j\big)$.

Putting everything together, for a set of observed context points C, the generative process of NDPs is given by the expression below, where we emphasise once again that $l(t_i)$ also implicitly depends on $l(t_0)$ and $d$.

$$p\big(y_{1:n}, l(t_0), d|t_{1:n}, C\big) = p\big(l(t_0)|C\big)p(d|C)\prod_{i=1}^{n} p\big(y_i|g(l(t_i), t_i)\big), \tag{4}$$

We remark that **NDPs generalise NPs defined over time**. If the latent NODE learns the trivial velocity $f_\theta(l(t), d, t) = 0$, the random state $L(t) = L_0$ remains constant at all times $t$. In this case, the distribution over functions is directly determined by $L_0 \sim p(l(t_0)|C)$, which substitutes the random variable $Z$ from a regular NP. For greater flexibility, the control signal $d$ can also be supplied

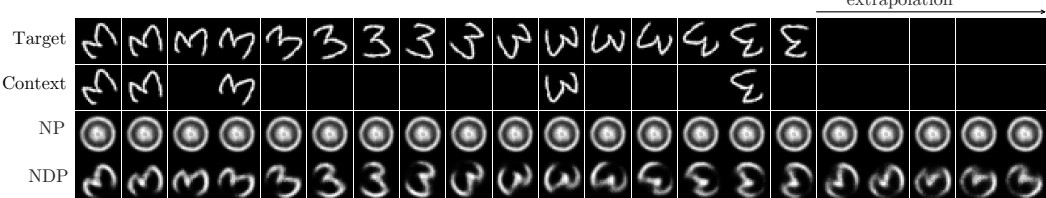

Figure 3: Predictions on the test set of Variable Rotating MNIST. NDP is able to extrapolate beyond the training time range whereas NP cannot even learn to reconstruct the digit.

to the decoder $g(\boldsymbol{l}(t), \boldsymbol{d}, t)$. This shows that, in principle, NDPs are at least as expressive as NPs. Therefore, NDPs could be a sensible choice even in applications where the time-series are not solely determined by some underlying dynamics, but are also influenced by other generative factors.

## 2.2  Learning and Inference

Since the true posterior is intractable because of the highly non-linear generative process, the model is trained using an amortised variational inference procedure. The variational lower-bound on the probability of the target values given the known context $\log p(y_\mathrm{T}|t_\mathrm{T}, y_\mathrm{C})$ is as follows:

$$\mathbb{E}_{q\big(\boldsymbol{l}(t_0), \boldsymbol{d}|t_\mathrm{T}, y_\mathrm{T}\big)}\left[\sum_{i \in I_\mathrm{T}} \log p(\boldsymbol{y}_i|\boldsymbol{l}(t_0), \boldsymbol{d}, t_i) + \log \frac{q_L(\boldsymbol{l}(t_0)|t_\mathrm{C}, y_\mathrm{C})}{q_L(\boldsymbol{l}(t_0)|t_\mathrm{T}, y_\mathrm{T})} + \log \frac{q_D(\boldsymbol{d}|t_\mathrm{C}, y_\mathrm{C})}{q_D(\boldsymbol{d}|t_\mathrm{T}, y_\mathrm{T})}\right], \quad (5)$$

where $q_L$, $q_D$ give the variational posteriors (the encoders described in Section 2.1). The full derivation can be found in Appendix E. We use the reparametrisation trick to backpropagate the gradients of this loss. During training, we sample random contexts of different sizes to allow the model to become sensitive to the size of the context and the location of its points. We train using mini-batches composed of multiple contexts. For that, we use an extended ODE that concatenates the independent ODE states of each sample in the batch and integrates over the union of all the times in the batch [9]. Pseudo-code for this training procedure is also given in Appendix F.

## 3  Experiments

To test our model on high-dimensional time-series with latent dynamics, we consider the rotating MNIST digits [1, 12]. In the original task, samples of digit "3" start upright and rotate once over 16 frames $(= 360°s^{-1})$ (i.e. constant angular velocity, zero angular shift). However, since we are interested in time-series with variable latent dynamics and increased variability in the initial conditions as in our formal problem statement, we consider a more challenging version of the task. In our adaptation, the angular velocity varies between samples in the range $(360° \pm 60°)s^{-1}$ and each sample starts at a random initial rotation. To induce some irregularity in each time-series in the training dataset, we remove five randomly chosen time-steps (excluding the initial time $t_0$) from each time-series. Overall, we generate a dataset with $1,000$ training time-series, 100 validation time-series and 200 test time-series, each using disjoint combinations of different calligraphic styles and dynamics. We compare NPs and NDPs using identical convolutional networks for encoding the images in the context. We assume that the initial image $y_0$ (i.e. the image at $t_0$) is always present in the context. As such, for NDPs, we compute the distribution of $L_0$ purely by encoding $y_0$ and disregarding the other samples in the context, as described in Section 2. We train the NP for $500$ epochs and use the validation set error to checkpoint the best model for testing. We follow a similar procedure for the NDP model but, due to the additional computational load introduced by the integration operation, only train for 50 epochs.

In Figure 3, we include the predictions offered by the two models on a time-series from the test dataset, which was not seen in training by either of the models. Despite the lower number of epochs they are trained for, NDPs are able to interpolate and even extrapolate on the variable velocity MNIST dataset, while also accurately capturing the calligraphic style of the digit. NPs struggle on this challenging task and are unable to produce anything resembling the digits. In order to better understand this wide performance gap, we also train in Appendix J.3 the exact same models on the easier Rotating MNIST task from Çağatay Yıldız et al. [12] where the angular velocity and initial rotation are constant. In

this setting, the two models perform similarly since the NP model can rely on simple interpolations without learning any dynamics.

## 4 Conclusion

We introduce Neural ODE Processes (NDPs), a new class of stochastic processes suitable for modelling data-adaptive stochastic dynamics. First, NDPs tackle the two main problems faced by Neural ODEs applied to dynamics-governed time series: adaptability to incoming data points and uncertainty in the underlying dynamics when the data is sparse and, potentially, irregularly sampled. Second, they add an explicit treatment of time as an additional inductive bias inside Neural Processes. To do so, NDPs include a probabilistic ODE as an additional encoded structure, thereby incorporating the assumption that the time-series is the direct or latent manifestation of an underlying ODE. Furthermore, NDPs maintain the scalability of NPs to large inputs. We evaluate our model on a high-dimensional problem – the rotating MNIST digits. Our method exhibits superior training performance when compared with NPs, yielding a lower loss in fewer iterations. We find that when there is a fundamental ODE governing the dynamics, NDPs perform well.

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

## A    Running Time Complexity

For a model with $n$ context points and $m$ target points, an NP has running time complexity $O(n+m)$, since the model only has to encode each context point and decode each target point. However, a Neural ODE Process has added complexity due to the integration process. Firstly, the integration itself has runtime complexity $O(\text{NFE})$, where NFE is the number of function evaluations. In turn, the worst-case NFE depends on the minimum step size $\delta$ the ODE solver has to use and the maximum integration time we are interested in, which we denote by $\Delta t_{\max}$. Secondly, for settings where the target times are not already ordered, an additional $O\big(m\log(m)\big)$ term is added for sorting them. This ordering is required by the ODE solver.

Therefore, given that $m \geq n$ and assuming a constant $\Delta t_{\max}$ exists, the worst-case complexity of NDPs is $O\big(m\log(m)\big)$. For applications where the times are already sorted (e.g. real-time applications), the complexity falls back to the original $O(n+m)$. In either case, NDPs scale well with the size of the input. We note, however, that the integration steps $\Delta t_{\max}/\delta$ could result in a very large constant, hidden by the big-$O$ notation. Nonetheless, modern ODE solvers use adaptive step sizes that adjust to the data that has been supplied and this should alleviate this problem. In our experiments, when sorting is used, we notice the NDP models are between 1 and 1.5 orders of magnitude slower to train than NPs in terms of wall-clock time. At the same time, this limitation of the method is traded-off by a significantly faster loss decay per epoch and superior final performance. We provide a table of time ratios from our 1D experiments, from Appendix C.1, in Appendix G.

## B    Model Variations

Here we present the different ways to implement the model. The majority of the variation is in the architecture of the decoder. However, it is possible to vary the encoder such that $f_e((t_i^{\text{C}}, \boldsymbol{y}_i^{\text{C}}))$ can be a multi-layer-perceptron, or additionally contain convolutions.

**Neural ODE Process (NDP)**    In this setup the decoder is an arbitrary function $g(\boldsymbol{l}(t_i^{\text{T}}), \boldsymbol{d}, t_i^{\text{T}})$ of the latent position at the time of interest, the control signal, and time. This type of model is particularly suitable for high-dimensional time-series where the dynamics are fundamentally latent. The inclusion of $\boldsymbol{d}$ in the decoder offers the model additional flexibility and makes it a good default choice for most tasks.

**Second Order Neural ODE Process (ND2P)**    This variation has the same decoder architecture as NDP, however the latent ODE evolves according to a second order ODE. The latent state, $\boldsymbol{l}$, is split into a "position", $\boldsymbol{l}_1$ and "velocity", $\boldsymbol{l}_2$, with $\dot{\boldsymbol{l}}_1 = \boldsymbol{l}_2$ and $\dot{\boldsymbol{l}}_2 = f_\theta(\boldsymbol{l}_1, \boldsymbol{l}_2, \boldsymbol{d}, t)$. This model is designed for time-series where the dynamics are second-order, which is often the case for physical systems [8, 12].

**NDP Latent-Only (NDP-L)**    The decoder is a linear transformation of the latent state $g(\boldsymbol{l}(t_i^{\text{T}})) = \boldsymbol{W}(\boldsymbol{l}(t_i^{\text{T}})) + \boldsymbol{b}$. This model is suitable for the setting when the dynamics are fully observed (i.e. they are not latent) and, therefore, do not require any decoding. This would be suitable for simple functions generated by ODEs, for example, sines and exponentials. This decoder implicitly contains information about time and $\boldsymbol{d}$ because the ODE evolution depends on these variables as described in Equation 3.

**ND2P Latent-Only (ND2P-L)**    This model combines the assumption of second-order dynamics with the idea that the dynamics are fully observed. The decoder is a linear layer of the latent state as in NDP-L and the phase space dynamics are constrained as in ND2P.

## C    Low-dimensional experiments

To test the proposed advantages of NDPs we carried out various experiments on time series data. For the low-dimensional experiments in Sections C.1 and C.2, we use an MLP architecture for the encoder and decoder. For the high-dimensional experiments in Section 3, we use a convolutional architecture for both. We train the models using RMSprop [10] with learning rate $1 \times 10^{-3}$. Additional model and task details can be found in Appendices I and J, respectively.

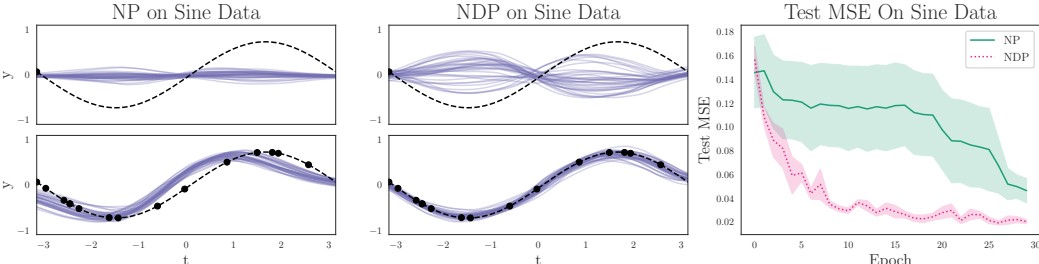

Figure 4: We present example posteriors of trained models and the loss during training of the NP and NDP models for the sine data. We find that NDPs are able to produce a greater range of functions when a single context point is provided, and a sharper, better targeted range as more points in the time series are observed. Quantitatively, NDPs train to a lower loss in fewer epochs, as may be expected for functions that are generated by ODEs. Both models were trained for 30 epochs.

## C.1   One Dimensional Regression

We begin with a set of 1D regression tasks of differing complexity—sine waves, exponentials, straight lines and damped oscillators—that can be described by ODEs. For each task, the functions are determined by a set of parameters (amplitude, shift, etc.) with pre-defined ranges. To generate the distribution over functions, we sample these parameters from a uniform distribution over their respective ranges. We use 490 time-series for training and evaluate on 10 separate test time-series. Each series contains 100 points. We repeat this procedure across 5 different random seeds to compute the standard error. Additional details can be found in Appendix J.1.

The left and middle panels of Figure 4 show how NPs and NDPs adapt on the sine task to incoming data points. When a single data-point has been supplied, NPs have incorrectly collapsed the distribution over functions to a set of almost horizontal lines. NDPs, on the other hand, are able to produce a wide range of possible trajectories. Even when a large number of points have been supplied, the NP posterior does not converge on a good fit, whereas NDPs correctly capture the true sine curve. In the right panel of Figure 4, we show the test-set MSE as a function of the training epoch. It can be seen that NDPs train in fewer iterations to a lower test loss despite having approximately 10% fewer parameters than NPs. We conducted an ablation study, training all model variants on all the 1D datasets, with final test MSE losses provided in Table 1 and training plots in Appendix J.1.

We find that NDPs either strongly outperform NPs (sine, linear), or their standard errors overlap (exponential, oscillators). For the exponential and harmonic oscillator tasks, where the models perform similarly, many points are close to zero in each example and as such it is possible to achieve a low MSE score by producing outputs that are also around zero. In contrast, the sine and linear datasets have a significant variation in the $y$-values over the range, and we observe that NPs perform considerably worse than the NDP models on these tasks.

The difference between NDP and the best of the other model variants is not significant across the set of tasks. As such, we consider only NDPs for the remainder of the paper as this is the least constrained model version: they have unrestricted latent phase-space dynamics, unlike the second-order counterparts, and a more expressive decoder architecture, unlike the latent-only variants. In addition, NDPs train in a faster wall clock time than the other variants, as shown in Appendix G.

**Active Learning**   We perform an active learning experiment on the sines dataset to evaluate both the uncertainty estimates produced by the models and how well they adapt to new information. Provided with an initial context point, additional points are greedily queried according to the model uncertainty. Higher quality uncertainty estimation and better adaptation will result in more information being acquired at each step, and therefore a faster and greater reduction in error. As shown in Figure 5, NDPs also perform better in this setting.

## C.2   Predator-Prey Dynamics

The Lotka-Volterra Equations are used to model the dynamics of a two species system, where one species predates on the other. The populations of the prey, $u$, and the predator, $v$, are given by

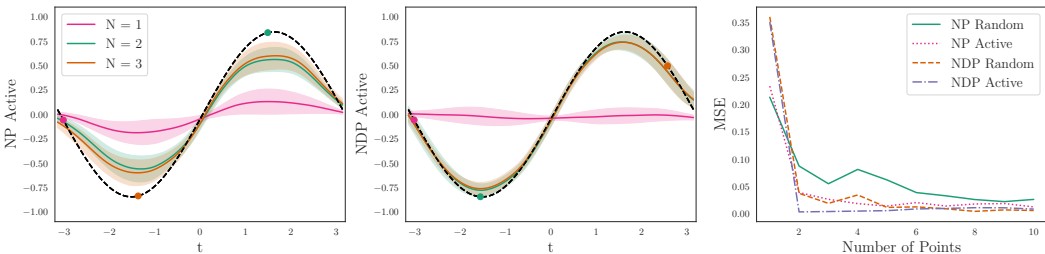

Figure 5: Active learning on the sines dataset. *Left:* NPs querying the points of highest uncertainty. *Middle:* NDPs querying the points of highest uncertainty, qualitatively it outperforms NPs. *Right:* MSE plots of four different querying regimes, NPs and NDPs looking actively and randomly, NDP Active decreases the MSE the fastest.

Table 1: Final MSE Loss on 1D regression tasks with standard error (lower is better). **Bold** indicates that the model performance is within error at the 95% confidence level, with underline indicating the best estimate for top-performer. NPs perform similarly to NDPs and their variants on the exponential and oscillator tasks, where $y$-values are close to zero. For the sine and linear tasks, where $y$ values vary significantly over the time range, NPs perform worse than NDPs and their variants.

| | MSE $\times 10^{-2}$ | | | |
| Model | Sine | Linear | Exponential | Oscillators |
|---|---|---|---|---|
| NP | $5.93 \pm 0.96$ | $5.85 \pm 0.70$ | $\mathbf{0.29 \pm 0.03}$ | $\mathbf{0.64 \pm 0.06}$ |
| NDP | $\mathbf{2.09 \pm 0.12}$ | $\mathbf{3.76 \pm 0.32}$ | $\mathbf{0.31 \pm 0.08}$ | $\mathbf{0.72 \pm 0.08}$ |
| ND2P | $2.75 \pm 0.19$ | $\mathbf{4.37 \pm 1.14}$ | $\underline{\mathbf{0.25 \pm 0.04}}$ | $\mathbf{0.55 \pm 0.03}$ |
| NDP-L | $\mathbf{2.51 \pm 0.24}$ | $4.77 \pm 0.67$ | $0.40 \pm 0.04$ | $0.72 \pm 0.04$ |
| ND2P-L | $\mathbf{2.64 \pm 0.30}$ | $\underline{\mathbf{3.16 \pm 0.46}}$ | $0.39 \pm 0.05$ | $0.66 \pm 0.03$ |

the differential equations $\dot{u} = \alpha u - \beta uv, \dot{v} = \delta uv - \gamma v$, for positive real parameters, $\alpha, \beta, \delta, \gamma$. Intuitively, when prey is plentiful, the predator population increases $(+\delta uv)$, and when there are many predators, the prey population falls $(-\beta uv)$. The populations exhibit periodic behaviour, with the phase-space orbit determined by the conserved quantity $V = \delta u - \gamma \ln(u) + \beta v - \alpha \ln(v)$. Thus for any predator-prey system there exists a range of stable functions describing the dynamics, with any particular realisation being determined by the initial conditions, $(u_0, v_0)$. We consider the system $(\alpha, \beta, \gamma, \delta) = (^2/_3, ^4/_3, 1, 1)$.

We generate sample time-series from the Lotka Volterra system by considering different starting configurations; $(u_0, v_0) = (2E, E)$, where $E$ is sampled from a uniform distribution in the range $(0.25, 1.0)$. The training set consists of 40 such samples, with a further 10 samples forming the test set. As before, each time series consists of 100 time samples and we evaluate across 5 different random seeds to obtain a standard error.

We find that NDPs are able to train in fewer epochs to a lower loss (Appendix J.2). We record final test MSEs $(\times 10^{-2})$ at $44 \pm 4$ for the NPs and $15 \pm 2$ for the NDPs. As in the 1D tasks, NDPs perform better despite having a representation $r$ and context $z$ with lower dimensionality, leading to 10% fewer parameters than NPs. Figure 6 presents these advantages for a single time series.

## D   Stochastic Process Proofs

Before giving the proofs, we state the following important Lemma.

**Lemma D.1.** *As in NPs, the decoder output $g(\boldsymbol{l}(t), t)$ can be seen as a function $\mathcal{F}(t)$ for a given fixed $\boldsymbol{l}(t_0)$ and $\boldsymbol{d}$.*

*Proof.* This follows directly from the fact that $\boldsymbol{l}(t) = \boldsymbol{l}(t_0) + \int_{t_0}^{\mathrm{T}} f_\theta(\boldsymbol{l}(t), t, \boldsymbol{d})dt$ can be seen as a function of $t$ and that the integration process is deterministic for a given pair $\boldsymbol{l}(t_0)$ and $\boldsymbol{d}$ (i.e. for fixed initial conditions and control). $\square$

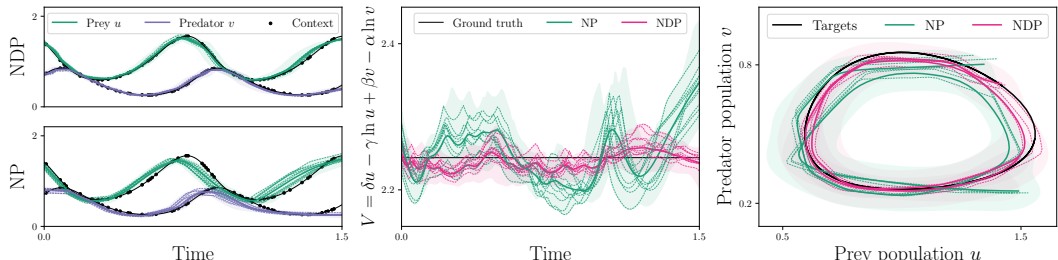

Figure 6: NPs and NDPs on the Lotka-Volterra task. Black is used for targets or ground truth, solid lines for mean predictions over 50 samples, and dashed lines for sample trajectories. In the *left* and *middle* plots, the shaded regions show the min-max range over 50 samples, in the *right* plot the shaded region was produced using kernel density estimation. *Left:* NPs are less able to model the dynamics, diverging from the ground truth even in regions with dense context sampling, whereas the NDP is both more accurate and varies more appropriately. *Middle:* Plotting the theoretically conserved quantity $V$ better exposes how the models deviate from the ground truth *Right:* In phase space $(u, v)$ the NDP is more clearly seen to better track the ground truth.

**Proposition D.1** *NDPs satisfy the exchangeability condition.*

*Proof.* This follows directly from Lemma D.1, since any permutation on $t_{1:n}$ would automatically act on $\mathcal{F}_{1:n}$ and consequently on $p(\boldsymbol{y}_{1:n}, \boldsymbol{l}(t_0), \boldsymbol{d}|t_{1:n})$, for any given $\boldsymbol{l}(t_0), \boldsymbol{d}$.  □

**Proposition D.2** *NDPs satisfy the consistency condition.*

*Proof.* Based on Lemma D.1 we can write the joint distribution (similarly to a regular NP) as follows:

$$\rho_{t_{1:n}}(y_{1:n}) = \int p(\mathcal{F}) \prod_{i=1}^{n} p(\boldsymbol{y}_i|\mathcal{F}(t_i))d\mathcal{F}. \tag{6}$$

Because the density of any $\boldsymbol{y}_i$ depends only on the corresponding $t_i$, integrating out any subset of $\boldsymbol{y}_{1:n}$ gives the joint distribution of the remaining random variables in the sequence. Thus, consistency is guaranteed.  □

# E   ELBO Derivation

As noted in Lemma D.1, the joint probability $p(\boldsymbol{y}, \boldsymbol{l}(t_0), \boldsymbol{d}|t) = p(\boldsymbol{l}(t_0))p(\boldsymbol{d})p(\boldsymbol{y}|g(\boldsymbol{l}(t), \boldsymbol{d}, t))$ can still be seen as a function that depends only on $t$, since the ODE integration process is deterministic for a given $\boldsymbol{l}(t_0)$ and $\boldsymbol{d}$. Therefore, the ELBO derivation proceeds as usual [5]. For convenience, let $\boldsymbol{z} = (\boldsymbol{l}(t_0), \boldsymbol{d})$ denote the concatenation of the two latent vectors and $q(\boldsymbol{z}) = q_L(\boldsymbol{l}(t_0))q_D(\boldsymbol{d})$. First, we derive the ELBO for $\log p(y_\mathrm{T}|t_\mathrm{T})$.

$$\log p(y_\mathrm{T}|t_\mathrm{T}) = D_\mathrm{KL}\big(q(\boldsymbol{z}|t_\mathrm{T}, y_\mathrm{T})\|p(\boldsymbol{z}|t_\mathrm{T}, y_\mathrm{T})\big) + \mathcal{L}_\mathrm{ELBO} \tag{7}$$

$$\geq \mathcal{L}_\mathrm{ELBO} = \mathbb{E}_{q(\boldsymbol{z}|t_\mathrm{T}, y_\mathrm{T})}\big[-\log q(\boldsymbol{z}|t_\mathrm{T}, y_\mathrm{T}) + \log p(y_\mathrm{T}, \boldsymbol{z}|t_\mathrm{T})\big] \tag{8}$$

$$= -\mathbb{E}_{q(\boldsymbol{z}|t_\mathrm{T}, y_\mathrm{T})}\log q(\boldsymbol{z}|t_\mathrm{T}, y_\mathrm{T}) + \mathbb{E}_{q(\boldsymbol{z}|t_\mathrm{T}, y_\mathrm{T})}\big[\log p(\boldsymbol{z}) + \log p(y_\mathrm{T}|t_\mathrm{T}, \boldsymbol{z})\big] \tag{9}$$

$$= \mathbb{E}_{q(\boldsymbol{z}|t_\mathrm{T}, y_\mathrm{T})}\left[\sum_{i \in I_\mathrm{T}} \log p(\boldsymbol{y}_i|\boldsymbol{z}, t_i) + \log \frac{p(\boldsymbol{z})}{q(\boldsymbol{z}|t_\mathrm{T}, y_\mathrm{T})}\right] \tag{10}$$

Noting that at training time, we want to maximise $\log p(y_\mathrm{T}|t_\mathrm{T}, y_\mathrm{C})$. Using the derivation above, we obtain a similar lower-bound, but with a new prior $p(z|t_\mathrm{C}, y_\mathrm{C})$, updated to reflect the additional information supplied by the context.

$$\log p(y_\mathrm{T}|t_\mathrm{T}, y_\mathrm{C}) \geq \mathbb{E}_{q(\boldsymbol{z}|t_\mathrm{T}, y_\mathrm{T})}\left[\sum_{i \in I_\mathrm{T}} \log p(\boldsymbol{y}_i|\boldsymbol{z}, t_i) + \log \frac{p(\boldsymbol{z}|t_\mathrm{C}, y_\mathrm{C})}{q(\boldsymbol{z}|t_\mathrm{T}, y_\mathrm{T})}\right] \tag{11}$$

If we approximate the true $p(z|t_\text{C}, y_\text{C})$ with the variational posterior, this takes the final form

$$\log p(y_\text{T}|t_\text{T}, y_\text{C}) \geq \mathbb{E}_{q(\boldsymbol{z}|t_\text{T}, y_\text{T})}\left[ \sum_{i \in I_\text{T}} \log p(\boldsymbol{y}_i|\boldsymbol{z}, t_i) + \log \frac{q(\boldsymbol{z}|t_\text{C}, y_\text{C})}{q(\boldsymbol{z}|t_\text{T}, y_\text{T})} \right] \qquad (12)$$

Splitting $\boldsymbol{z} = (\boldsymbol{l}(t_0), \boldsymbol{d})$ back into its constituent parts, we obtain the loss function

$$\mathbb{E}_{q\left(\boldsymbol{l}(t_0), \boldsymbol{d}|t_\text{T}, y_\text{T}\right)}\left[ \sum_{i \in I_\text{T}} \log p(\boldsymbol{y}_i|\boldsymbol{l}(t_0), \boldsymbol{d}, t_i) + \log \frac{q_L(\boldsymbol{l}(t_0)|t_\text{C}, y_\text{C})}{q_L(\boldsymbol{l}(t_0)|t_\text{T}, y_\text{T})} + \log \frac{q_D(\boldsymbol{d}|t_\text{C}, y_\text{C})}{q_D(\boldsymbol{d}|t_\text{T}, y_\text{T})} \right]. \qquad (13)$$

## F   Learning and Inference Procedure

We include below the pseudocode for training NDPs. For clarity of exposition, we give code for a single time-series. However, in practice, we batch all the operations in lines $6 - 15$.

---

**Algorithm 1:** Learning and Inference in Neural ODE Processes

---

**Input**   : A dataset of time-series $\{\boldsymbol{X}_k\}$, $k \leq K$, where $K$ is the total number of time-series

1  Initialise NDP model with parameters $\theta$
2  Let $m$ be the number of context points and $n$ the number of extra target points
3  **for** $i \leftarrow 0$ **to** *training_steps* **do**
4  $\quad$ Sample $m$ from $\text{U}[min\_context\_points, max\_context\_points]$
5  $\quad$ Sample $n$ from $\text{U}[min\_extra\_target\_points, max\_extra\_target\_points]$
6  $\quad$ Uniformly sample a time-series $\boldsymbol{X}_k$
7  $\quad$ Uniformly sample from $\boldsymbol{X}_k$ the target points $\text{T} = (t_\text{T}, y_\text{T})$, where $t_\text{T}$ is the time batch with
$\quad\quad$ shape $(m + n, 1)$ and $y_\text{T}$ is the corresponding outputs batch with shape $(m + n, \dim(\boldsymbol{y}))$
8  $\quad$ Extract the (unordered) context set $\text{C} = \text{T}[0 : m]$
9  $\quad$ Compute $q(\boldsymbol{l}(t_0), \boldsymbol{d}|\text{C})$ using the variational encoder
10 $\quad$ Compute $q(\boldsymbol{l}(t_0), \boldsymbol{d}|\text{T})$ using the variational encoder
$\quad$ // During training, we sample from $q(\boldsymbol{l}(t_0), \boldsymbol{d}|\text{T})$
11 $\quad$ Sample $\boldsymbol{l}(t_0), \boldsymbol{d}$ from $q(\boldsymbol{l}(t_0), \boldsymbol{d}|\text{T})$
12 $\quad$ Integrate to compute $\boldsymbol{l}(t)$ as in Equation 3 for all times $t \in t_\text{T}$
13 $\quad$ **foreach** *time* $t \in t_\text{T}$ **do**
14 $\quad\quad$ Use decoder to compute $p(\boldsymbol{y}(t)|g(\boldsymbol{l}(t)), t)$
15 $\quad$ Compute loss $\mathcal{L}_\text{ELBO}$ based on Equation 5
16 $\quad$ $\theta \leftarrow \theta - \alpha\nabla_\theta\mathcal{L}_\text{ELBO}$

---

It is worth highlighting that during training we sample $\boldsymbol{l}(t_0), \boldsymbol{d}$ from the target-conditioned posterior, rather than the context-conditioned posterior. In contrast, at inference time we sample from the context-conditioned posterior.

## G   Wall Clock Training Times

To explore the additional term in the runtime given in Section A, we record the wall clock time for each model to train for 30 epochs on the 1D synthetic datasets, over 5 seeds. Then we take the ratio of a given model and the NP. The experiments were run on an *Nvidia Titan XP*. The results can be seen in Table 2.

## H   Size of Latent ODE

To investigate how many dimensions the ODE $\boldsymbol{l}$ should have, we carry out an ablation study, looking at the performance on the 1D sine dataset. We train models with $\boldsymbol{l}$-dimension $\{1, 2, 5, 10, 15, 20\}$ for 30 epochs. Figure 7 shows training plots for $\dim(\boldsymbol{l}) = \{1, 2, 10, 20\}$, and final MSE values are given in Table 3.

We see that when $\dim(\boldsymbol{l}) = 1$, NDPs are slow to train and require more epochs. This is because sine curves are second-order ODEs, and at least two dimensions are required to learn second-order

Table 2: Table of ratios, of Neural ODE Process and Neural Process training times on different 1D synthetic datasets. We see that NDP/NP is the lowest (i.e. fastest) in each case.

| Time Ratios | Sine | Exponential | Linear | Oscillators |
|---|---|---|---|---|
| NDP/NP | **22.1 ± 0.9** | **23.6 ± 0.9** | **10.9 ± 1.4** | **22.2 ± 2.3** |
| ND2P/NP | 55.2 ± 6.3 | 32.4 ± 1.5 | 14.2 ± 0.3 | 35.8 ± 0.7 |
| NDP-L/NP | 55.2 ± 6.2 | 47.5 ± 18.0 | 14.7 ± 1.5 | 25.3 ± 0.5 |
| ND2P-L/NP | 43.7 ± 1.9 | 27.9 ± 1.1 | 15.1 ± 1.6 | 32.8 ± 1.1 |
| NP Training Time /s | 22.4 ± 0.2 | 45.5 ± 0.3 | 100.9 ± 0.3 | 23.2 ± 0.4 |

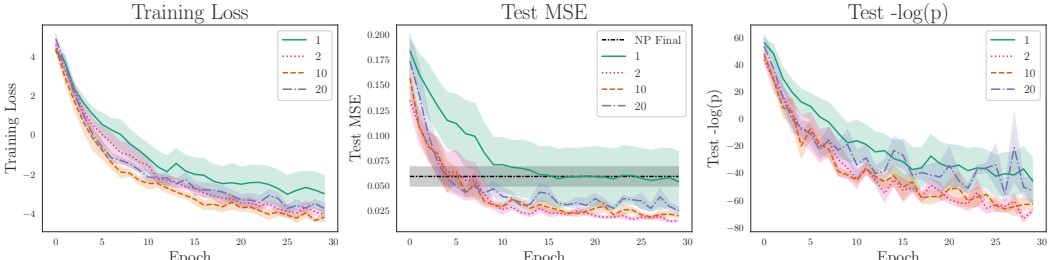

Figure 7: Training plots of NDP with ODEs of different sizes, training on the sine dataset. We see that for $\dim(\boldsymbol{l}) = 1$, the model trains slowly, as would be expected for a sine curve where at least 2 dimensions are needed to learn second order and test performance is close to the standard NP. The other models train at approximately the same rate.

dynamics (one for the position and one for the velocity). When $\dim(\boldsymbol{l}) = 1$, NDPs perform similarly to NPs, which is expected when the latent ODE is unable to capture the underlying dynamics. We then see that for all other dimensions, NDPs train at approximately the same rate (over epochs) and have similar final MSE scores. As the dimension increases beyond 10, the test MSE increases, indicating overfitting.

# I  Architectural Details

For the experiments with low dimensionality (1D, 2D), the architectural details are as follows:

- **Encoder**: $[t_i, y_i] \to \boldsymbol{r}_i$: Multilayer Perceptron, 2 hidden layers, ReLU activations.
- **Aggregator**: $\boldsymbol{r}_{1:n} \to \boldsymbol{r}$: Taking the mean.
- **Representation to Hidden**: $\boldsymbol{r} \to \boldsymbol{h}$: One linear layer followed by ReLU.
- **Hidden to $L_0$ Mean**: $\boldsymbol{h} \to \mu_L$: One linear layer.
- **Hidden to $L_0$ Variance**: $\boldsymbol{h} \to \sigma_L$: One linear layer, followed by sigmoid, multiplied by 0.9 add 0.1, i.e. $\sigma_L = 0.1 + 0.9 \times \text{sigmoid}(\boldsymbol{W}\boldsymbol{h} + \boldsymbol{b})$.
- **Hidden to $D$ Mean**: $\boldsymbol{h} \to \mu_D$: One linear layer.
- **Hidden to $D$ Variance**: $\boldsymbol{h} \to \sigma_D$: One linear layer, followed by sigmoid, multiplied by 0.9 add 0.1, i.e. $\sigma_D = 0.1 + 0.9 \times \text{sigmoid}(\boldsymbol{W}\boldsymbol{h} + \boldsymbol{b})$.
- **ODE Layers**: $[\boldsymbol{l}, \boldsymbol{d}, t] \to \dot{\boldsymbol{l}}$: Multilayer Perceptron, two hidden layers, $\tanh$ activations.
- **Decoder**: $g(\boldsymbol{l}(t_i^T), \boldsymbol{d}, t_i^T) \to y_i^T$, for the NDP model and ND2P described in section B, this function is a linear layer, acting on a concatenation of the latent state and a function of $\boldsymbol{l}(t_i^T)$, $\boldsymbol{d}$, and $t_i^T$. $g(\boldsymbol{l}(t_i^T), \boldsymbol{d}, t_i^T) = \boldsymbol{W}(\boldsymbol{l}(t_i^T) || h(\boldsymbol{l}(t_i^T), \boldsymbol{d}, t_i^T)) + \boldsymbol{b}$. Where $h$ is a Multilayer Perceptron with two hidden layers and ReLU activations.

For the high-dimensional experiments (Rotating MNIST).

Table 3: Final MSE values for NDPs training on the sine dataset with different sized ODEs, with NP performance included for reference. Peak performance is found when $\dim(\boldsymbol{l}) = 2$, which is to be expected as the true dynamics are 2-dimensional. For $\dim(\boldsymbol{l}) = 1$, the MSE is highest, and within error of the NP. Performance degrades with increasing $\dim(\boldsymbol{l})$, with overfitting becoming a problem for $\dim(\boldsymbol{l}) = 20$.

| $l$-dimension | MSE $\times 10^{-2}$ | Training Times /s |
|:---:|:---:|:---:|
| *NP* | *5.9 ± 0.9* | *22.4 ± 0.2* |
| 1 | 5.6 ± 1.3 | **299.7 ± 20.5** |
| 2 | **1.7 ± 0.1** | 413.8 ± 52.9 |
| 5 | 2.2 ± 0.2 | 414.8 ± 13.1 |
| 10 | 2.1 ± 0.1 | 496.7 ± 20.5 |
| 15 | 2.6 ± 0.2 | 618.0 ± 30.7 |
| 20 | 3.1 ± 0.3 | 652.0 ± 38.8 |

- **Encoder**: $[t_i, y_i] \to \boldsymbol{r}_i$: Convolutional Neural Network, 4 layers with 16, 32, 64, 128 channels respectively and kernel size of 5, stride 2. ReLU activations. Batch normalisation.

- **Aggregator**: $\boldsymbol{r}_{1:n} \to \boldsymbol{r}$: Taking the mean.

- **Representation to $D$ Hidden**: $\boldsymbol{r} \to \boldsymbol{h}_D$: One linear layer followed by ReLU.

- **$D$ Hidden to $D$ Mean**: $\boldsymbol{h}_D \to \mu_D$: One linear layer.

- **$D$ Hidden to $D$ Variance**: $\boldsymbol{h}_D \to \sigma_D$: One linear layer, followed by sigmoid, multiplied by 0.9 add 0.1, i.e. $\sigma_D = 0.1 + 0.9 \times \text{sigmoid}(\boldsymbol{W}\boldsymbol{h}_D + \boldsymbol{b})$.

- **$\boldsymbol{y}_0$ to $L_0$ Hidden**: $\boldsymbol{y}_0 \to \boldsymbol{h}_L$: Convolutional Neural Network, 4 layers with 16, 32, 64, 128 channels respectively and kernel size of 5, stride 2. ReLU activations. Batch normalisation.

- **$L_0$ Hidden to $L_0$ Mean**: $\boldsymbol{h}_L \to \mu_L$: One linear layer.

- **$L_0$ Hidden to $L_0$ Variance**: $\boldsymbol{h}_L \to \sigma_L$: One linear layer, followed by sigmoid, multiplied by 0.9 add 0.1, i.e. $\sigma_L = 0.1 + 0.9 \times \text{sigmoid}(\boldsymbol{W}\boldsymbol{h}_L + \boldsymbol{b})$.

- **ODE Layers**: $[\boldsymbol{l}, \boldsymbol{d}, t] \to \dot{\boldsymbol{l}}$: Multilayer Perceptron, two hidden layers, $\tanh$ activations.

- **Decoder**: $g(\boldsymbol{l}(t_i^{\mathrm{T}})) \to y_i^{\mathrm{T}}$: 1 linear layer followed by a 4 layer transposed Convolutional Neural Network with 32, 128, 64, 32 channels respectively. ReLU activations. Batch normalisation.

## J  Task Details and Additional Results

### J.1  One Dimensional Regression

We carried out an ablation study over model variations on various 1D synthetic tasks—sines, exponentials, straight lines and harmonic oscillators. Each task is based on some function described by a set of parameters that are sampled over to produce a distribution over functions. In every case, the parameters are sampled from uniform distributions. A trajectory example is formed by sampling from the parameter distributions and then sampling from that function at evenly spaced timestamps, $t$, over a fixed range to produce 100 data points $(t, y)$. We give the equations for these tasks in terms of their defining parameters and the ranges for these parameters in Table 4.

To test after each epoch, 10 random context points are taken, and then the mean-squared error and negative log probability are calculated over all the points (not just a subset of the target points). Each model was trained 5 times on each dataset (with different weight initialisation). We used a batch size of 5, with context size ranging from 1 to 10, and the extra target size ranging from 0 to 5.[2] The results are presented in Figure 8.

All models perform better than NPs, with fewer parameters (approximately 10% less). Because there are no significant differences between the different models, we use NDP in the remainder of

---

[2]As written in the problem statement in section 1, we make the context set a subset of the target set when training. So we define a context size range and an extra target size range for each task.

| Task | Form | a | b | t | # train | # test |
|------|------|---|---|---|---------|--------|
| Sines | $y = a\sin(t - b)$ | $(-1, 1)$ | $(-1/2, 1/2)$ | $(-\pi, \pi)$ | 490 | 10 |
| Exponentials | $y = {a}/{60} \times \exp(t - b)$ | $(-1, 1)$ | $(-1/2, 1/2)$ | $(-1, 4)$ | 490 | 10 |
| Straight lines | $y = at + b$ | $(-1, 1)$ | $(-1/2, 1/2)$ | $(0, 5)$ | 490 | 10 |
| Oscillators | $y = a\sin(t - b)\exp(-t/2)$ | $(-1, 1)$ | $(-1/2, 1/2)$ | $(0, 5)$ | 490 | 10 |

Table 4: Task details for 1D regression. $a$ and $b$ are sampled uniformly at random from the given ranges. $t$ is sampled at 100 regularly spaced intervals over the given range. 490 training examples and 10 test examples were used in every case.

the experiments, because it has the fewest model restrictions. The phase space dynamics are not restricted like its second-order variant, and the decoder has a more expressive architecture than the latent-only variants. It also trains the fastest in wall clock time seen in Appendix G.

### J.2 Lotka-Volterra System

To generate samples from the Lotka Volterra system, we sample different starting configurations, $(u_0, v_0) = (2E, E)$, where $E$ is sampled from a uniform distribution in the range (0.25, 1.0). We then evolve the Lotka Volterra system

$$\frac{du}{dt} = \alpha u - \beta uv, \qquad \frac{dv}{dt} = \delta uv - \gamma v. \tag{14}$$

using $(\alpha, \beta, \gamma, \delta) = (2/3, 4/3, 1, 1)$. This is evolved from $t = 0$ to $t = 15$ and then the times are rescaled by dividing by 10.

The training for the Lotka-Volterra system can be seen in Figure 9. This was taken across 5 seeds, with a training set of 40 trajectories, 10 test trajectories and batch size 5. We use a context size ranging from 1 to 100, and extra target size ranging from 0 to 45. The test context size was fixed at 90 query times. NDPs train slightly faster with lower loss, as expected.

### J.3 Rotating MNIST & Additional Results

To better understand what makes vanilla NPs fail on our Variable Rotating MNIST from Section 3, we train the exact same models on the simpler Rotating MNIST dataset [12]. In this dataset, all digits start in the same position and rotate with constant velocity. Additionally, the fourth rotation is removed from all the time-series in the training dataset. We follow the same training procedure as in Section 3.

We report in Figure 10 the predictions for the two models on a random time-series from the validation dataset. First, NPs and NDPs perform similarly well at interpolation and extrapolation within the time-interval used in training. As an exception but in agreement with the results from ODE$^2$VAE, NDPs produces a slightly better reconstruction for the fourth time step in the time-series. Second, neither model is able to extrapolate the dynamics beyond the time-range seen in training (i.e. the last five time-steps).

Overall, these observations suggest that for the simpler RotMNIST dataset, explicit modelling of the dynamics is not necessary and the tasks can be learnt easily by interpolating between the context points. And indeed, it seems that even NDPs, which should be able to learn solutions that extrapolate, collapse on these simpler solutions present in the parameter space, instead of properly learning the desired latent dynamics. A possible explanation is that the Variable Rotating MNIST dataset can be seen as an image augmentation process which makes the convolutional features to be approximately rotation equivariant. In this way, the NDP can also learn rotation dynamics in the spatial dimensions of the convolutional features.

Finally, in Figure 11, we plot the reconstructions of different digit styles on the test dataset of Variable Rotating MNIST. This confirms that NDPs are able to capture different calligraphic styles.

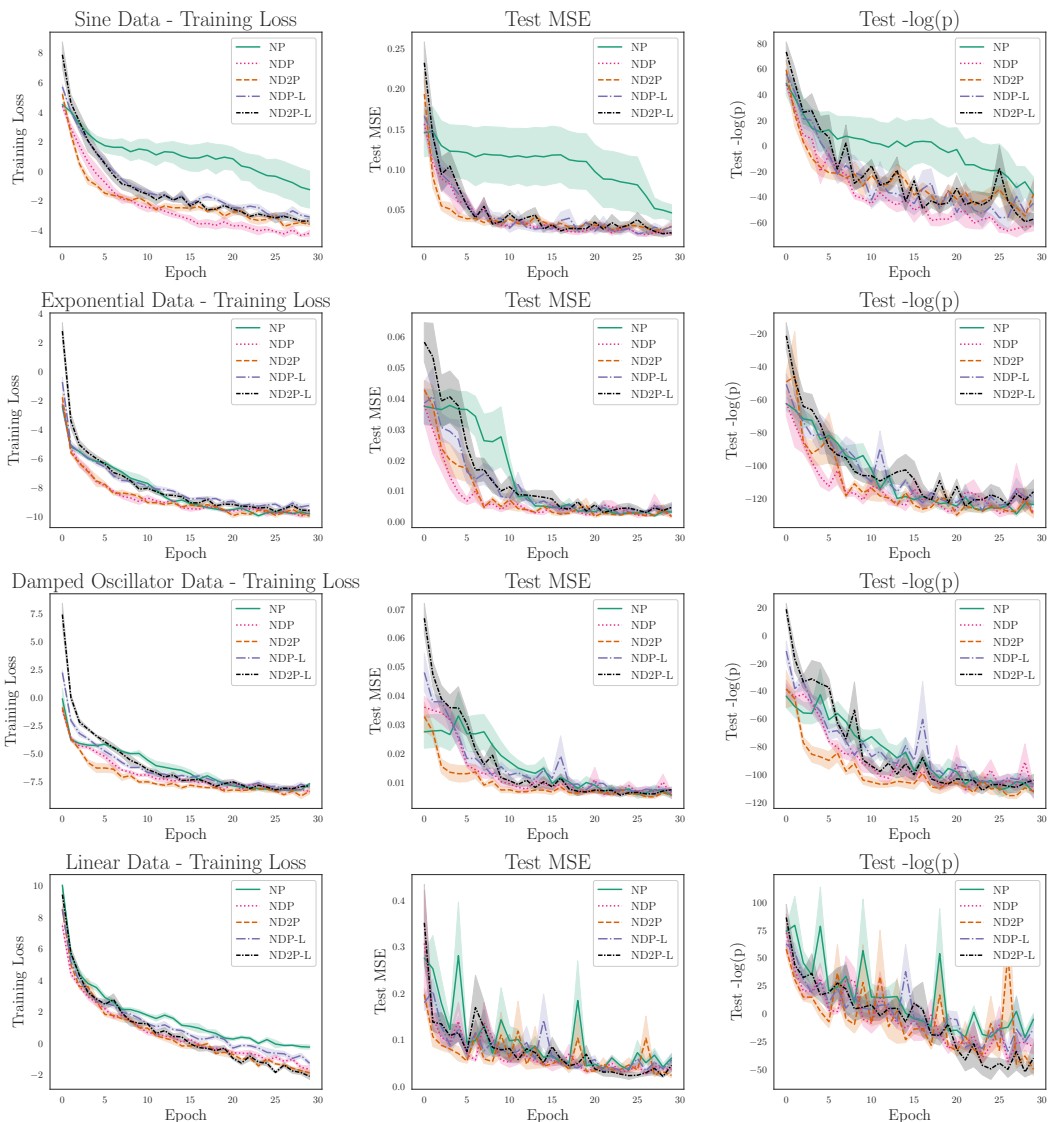

Figure 8: Training model variants on 1D synthetic datasets. NPs train slower in all cases. All Neural ODE Process variants train approximately at the same rate. With the latent-only variants performing slightly worse than the more expressive model variants. Additionally, ND2P performs slightly better than NDP on the damped oscillator and linear sets, because they are naturally easier to learn as second-order ODEs.

## J.4 Handwritten Characters

The `CharacterTrajectories` dataset consists of single-stroke handwritten digits recorded using an electronic tablet [3, 11]. The trajectories of the pen tip in two dimensions, $(x, y)$, are of varying length, with a force cut-off used to determine the start and end of a stroke. We consider a reduced dataset, containing only letters that were written in a single stroke, this disregards letters such as "f", "i" and "t". Whilst it is not obvious that character trajectories should follow an ODE, the related Neural Controlled Differential Equation (NCDEs) model has been applied successfully to this task [6]. We train with a training set with 49600 examples, a test set with 400 examples and a batch size of 200. We use a context size ranging between 1 and 100, an extra target size ranging between 0 and 100 and a fixed test context size of 20. We visualise the training of the models in Figure 12 and the models plotting posteriors in Figure 13.

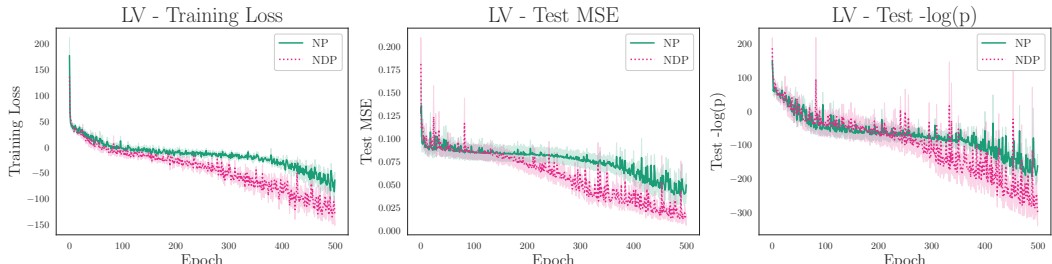

Figure 9: Training NP and NDP on the Lotka-Volterra equations. Due to the additional encoding structure of NDP, it can be seen that NDPs train in fewer iterations, to a lower loss than NPs.

extrapolation

Figure 10: Predictions on the simpler Rotating MNIST dataset. NPs are also able to perform well on this task, but NDPs are not able to extrapolate beyond the maximum training time.

We observe that NPs and NDPs are unable to successfully learn the time series as well as NCDEs. We record final test MSEs ($\times 10^{-1}$) at $4.6 \pm 0.1$ for NPs and a slightly lower $3.4 \pm 0.1$ for NDPs. We believe the reason is because handwritten digits do not follow an inherent ODE solution, especially given the diversity of handwriting styles for the same letter. We conjecture that Neural Controlled Differential Equations were able to perform well on this dataset due to the control process. Controlled ODEs follow the equation:

$$\boldsymbol{z}(T) = \boldsymbol{z}(t_0) + \int_{t_0}^{\mathrm{T}} f_\theta(\boldsymbol{z}(t), t) \frac{d\boldsymbol{X}(t)}{dt} dt, \qquad \boldsymbol{z}(t_0) = h_1(\boldsymbol{x}(t_0)), \qquad \hat{\boldsymbol{x}}(T) = h_2(\boldsymbol{z}(T)) \quad (15)$$

Where $\boldsymbol{X}(t)$ is the natural cubic spline through the observed points $\boldsymbol{x}(t)$. If the learnt $f_\theta$ is an identity operation, then the result returned will be the cubic spline through the observed points. Therefore, a controlled ODE can learn an identity with a small perturbation, which is easier to learn with the aid of a control process, rather than learning the entire ODE trajectory.

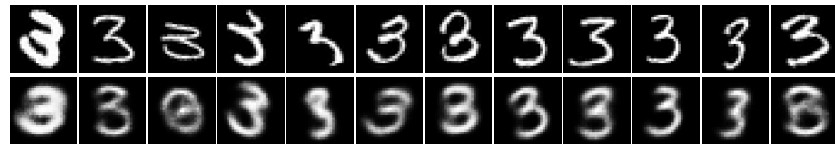

Figure 11: NDPs are able to capture different styles in the Variable Rotating MNIST dataset.

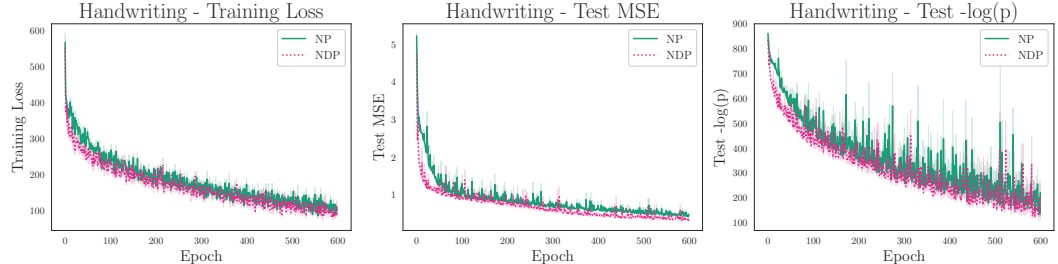

Figure 12: NPs and NDPs training on handwriting. NDPs perform slightly better, achieving a lower loss in fewer iterations. However this is a marginal improvement, and we believe it is down to significant diversity in the dataset, due to there being no fundamental differential equation for handwriting.

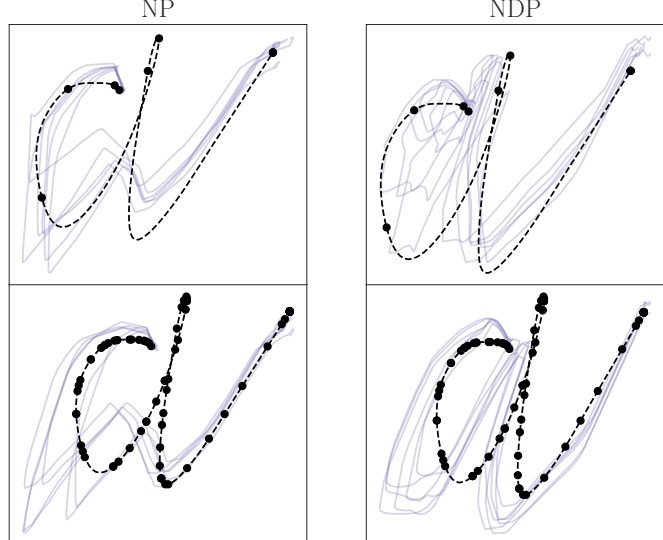

Figure 13: We test the models on drawing the letter "a" with varying numbers of context points. For a few context points, the trajectories are diverse and not entirely recognisable. As more context points are observed, the trajectories become less diverse and start approaching an "a". We expect that with more training, and tuning the hyperparameters, such as batch size, or the number of hidden layers this model would improve. Additionally, we observe that NDPs qualitatively outperform NPs on a small number and a large number of context points.

