# OpenReview forum: "Neural ODE Processes: A Short Summary"
_NeurIPS.cc/2021/Workshop/DLDE — DLDE Workshop -- NeurIPS 2021 Spotlight_

### Official Review · Reviewer_h2Jy · 2021-10-07
**Neural ODE Processes**

**Confidence:** 4

**Review:**

The authors use a combination of a graph-based model and deep neural networks to model a family of ODE functions that fit a set of samples (context points) plausibly, quantify the uncertainty and show that at times, their new model performs better under this uncertainty and using only a sparse set of context points, than previous work, as well as for more difficult problems such as their Variable Rotating MNIST dataset. The novelty lies mostly with the Neural Process model.

Since this paper has been recently published and is of high quality, I find no reason to critique it at length and only remark that at times, the notation used can be confusing. I feel that it could be simplified and made more consistent and that while reading the paper, I was sometimes unsure whether certain notations referred to the same thing and if so, why several different notations were used (or conversely, why different things would be denoted with very similar notation).

This is a good paper, of sound technical merit and the work is communicated and presented well.

**Score:**

4: Very good paper

---

### Official Review · Reviewer_yn97 · 2021-10-11

**Confidence:** 3

**Review:**

In their paper, the authors combine the concept of learned dynamics from Neural ODEs with the generative power of Neural Processes to achieve
a significant improvement on conditional time series inference. For this, the authors fit a latent Neural ODE-like model whose dynamics are dependent
on a given set of context points. They then use the resulting evolution of the latent variable to perform inference similarly to Neural Processes.
The authors show results on a time-series dataset of rotating MNIST digits indicating the power of an evolving latent variable.

The method seems to be novel and the authors show significant improvement over Neural Processes by combining them with concepts from Neural ODEs.
The paper is well written. However, sometimes the notation can be somewhat confusing which can be attributed to the limited length of submissions.

I think the presented method features a well thought-out combination of existing approaches which improves on significant flaws.

**Score:**

5: Excellent paper: should definitely be a contributed talk

---

### Official Review · Reviewer_RUEf · 2021-10-14
**Use of stochastic process to learn ODE of low and high dimensional system**

**Confidence:** 3

**Review:**

This paper introduces a new technique for learning over distributions of Neural ODEs in a streaming fashion to capture both low and high-dimensional system dynamics.  They are able to achieve significant improvement over Neural ODEs on MNIST.  The interesting contribution is the latent ODE in the network that is used to obtain a distribution over functions.  The supplementary material strongly supports the claims made in the paper and is clear to follow.  The implications of the work are potentially large since this work assumes a general, sparsely sampled time series of data.  The evaluation section is very thorough and compares with existing approaches and significantly outperforms them.

**Score:**

5: Excellent paper: should definitely be a contributed talk

---

### Decision · Program_Chairs · 2021-10-16

**Decision:**

Accept (Spotlight)

**Comment:**

All reviewers agree that this is an excellent piece of work, that should be accepted.

I would echo their concerns about notation/clarity. Various examples:

- The distinction between $\mathcal{D}$ and $\mathcal{D}'$ at the very start is not made clear until the very end of the paragraph (which is the point at which a latent space is introduced).
- The background on NP (Section 1) could have been introduced using t instead of x; this would not change the formulation of NP but would align more closely with the notation of the rest of the paper.
- More systemically, wordiness is often used where precise mathematical notation would have communicated the point more clearly (e.g. "Encoder and Aggregator", Section 2.1).
- Nonstandard notation (e.g. equation (2)) is also used systemically. (Not helped by such notations becoming more common in ML.) I'd challenge the authors to try communicating this in standard notation, i.e. as introduced in any first course on probability via measure theory.

Besides that, a few concerns of my own:

- It is not sufficiently clear to me what the distinction is between NDPs and the latent ODE model of Rubanova et al. 2019. Up to a few tweaks (choice of encoder, use of two latent states rather than one) they appear to be very similar. Given that latent ODEs are a familiar point of reference for many, an explicit comparison would be helpful.
- I think there may be some relatively fundamental limitations with the proposed architecture. In particular, the use of an ODE model presupposes drift-only dynamics. Would it be possible to model a diffusive process -- even a simple Brownian motion -- with an NDP? This links up with the topic of neural SDEs (where SDEs are of course one of the traditional ways to model random functions); are there interesting comparisons to be made with the latent SDE model of Li et al. 2020 or the SDE-GAN model of Kidger et al. 2021?